# Genetic background and immune response in paracoccidioidomycosis: A systematic review and meta-analysis of single nucleotide variants

Sanderson da Silva Coelho[1], Wellington Santos Fava[1], Eva Burger[2],
Ana Carla Pereira-Latini[3], Alessandra Pontillo[4], James Venturini[1]*

**1** School of Medicine, Universidade Federal de Mato Grosso do Sul, Campo Grande, Mato Grosso do Sul, Brazil, **2** Department of Microbiology and Immunology, Biomedical Sciences Institute, Federal University of Alfenas, Alfenas, Brazil, **3** Lauro de Souza Lima Institute, Bauru, Brazil, **4** Department of Immunology, Institute of Biomedical Sciences, University of São Paulo, São Paulo, Brazil

* james.venturini@ufms.br

## Abstract

Paracoccidioidomycosis (PCM) is a systemic fungal infection endemic to Latin America, especially Brazil, where it is considered a neglected occupational disease. Caused by *Paracoccidioides* spp., PCM presents a wide spectrum of clinical manifestations, ranging from localized to severe disseminated forms. This heterogeneity suggests that host-related factors, including genetic background, may influence disease development. Genetic susceptibility to infectious diseases is key to understanding differences in immune responses. In PCM, variants in immune-related genes such as cytokines and pattern recognition receptors may modulate susceptibility and disease progression. This study aimed to review the literature on the association between single nucleotide variants (SNVs) PCM susceptibility, severity, and clinical outcomes. A systematic review followed PRISMA guidelines, searching databases like MEDLINE (PubMed), Cochrane Library, LILACS, SciELO, Web of Science, and Google Scholar. Keywords related to "Paracoccidioidomycosis" and "SNVs" were used. The review included studies on SNVs and PCM susceptibility. The quality of the evidence was assessed with the Cochrane and Joanna Briggs Institute risk of bias tool. We also performed a meta-analysis of studies utilizing identical SNVs. This study is registered on PROSPERO, number CRD42025646417. Two SNVs, Interleukin 10 (*IL10*) and Vitamin D Receptor (*VDR*) showed a significant association in individual analyses, but none demonstrated a significant association in the meta-analysis. The main limitations discussed in the studies were insufficient sample size, population heterogeneity, and the composition of control groups. Although some SNVs, particularly in IL10 and VDR, showed significant associations with PCM susceptibility in individual studies, the evidence remains limited. The meta-analyses included only two studies per SNV, resulting in low statistical power and exploratory pooled estimates, largely reflecting small sample sizes, lack of replication, and methodological heterogeneity across studies.

**Data availability statement:** All relevant data are within the manuscript and its Supporting Information files.

**Funding:** This work was supported by the Coordenação de Aperfeiçoamento de Pessoal de Nível Superior (CAPES, Finance Code 001) to SSC and JV, and by the Universidade Federal de Mato Grosso do Sul (UFMS, Finance Code 001) to JV. The funders had no role in study design, data collection and analysis, decision to publish, or preparation of the manuscript.

**Competing interests:** The authors have declared that no competing interests exist.

## Author summary

Paracoccidioidomycosis is a fungal disease that affects mainly rural workers in Latin America, especially in Brazil, and remains largely neglected despite its social and health impact. People exposed to the fungus can develop different forms of the disease, from mild to severe, suggesting that individual biological factors play an important role. In this study, we asked whether genetic differences between people help explain who becomes ill and how severe the disease can be. We systematically reviewed published studies that investigated genetic variants in immune-related genes and their association with paracoccidioidomycosis. We also combined available data using meta-analysis when the same genetic variants were evaluated in more than one study. Although some studies suggested that variants in genes related to immune regulation and vitamin D signaling might influence susceptibility to the disease, these findings were not confirmed when the data were analyzed together. Our results show that current evidence linking human genetic variation to paracoccidioidomycosis is limited and inconsistent. Most studies included few participants and differed in their design, populations, and selection of control groups. We conclude that larger, better-designed studies are needed to clarify whether genetic factors truly influence the risk and clinical outcomes of this neglected fungal disease.

## Introduction

Paracoccidioidomycosis (PCM) is a systemic fungal infection endemic to Latin America, particularly Brazil, where it is considered a neglected tropical and occupational disease. It is caused by thermally dimorphic fungi of the genus *Paracoccidioides*, which includes *P. brasiliensis* species complex, composed of *P. brasiliensis sensu scrictu* (clade Sb1a-Sb1b), *P. americana* (PS2), *P. restrepiensis* (PS3), and *P. venezuelensis* (PS4), as well as the distinct species *P. lutzii* [1,2]. In addition to these cultivable species, the genus also includes non-cultivable species such as *P. ceti* and *P. lobogeorgii* [3]. PCM manifests in two main clinical forms: acute/subacute or chronic. The acute/subacute form is generally severe and predominantly affects children, adolescents, and young adults (occasionally up to age 30), progressing rapidly with widespread fungal dissemination to organs of the reticuloendothelial system, such as lymph nodes, liver, spleen, and bone marrow [4–6]. In contrast, the chronic form (around 80% of cases) mainly occurs in men (22:1 male-to-female ratio) aged 30–60. Pulmonary involvement is present in 90% of patients, followed by upper aerodigestive mucosa and skin. Disease severity ranges from mild to severe [4–6].

The complex fungus-host interaction in PCM reflects a dynamic and context-dependent equilibrium among fungal attributes, host immune responses, and environmental pressures [7]. Classical reviews have demonstrated that PCM pathogenesis is critically shaped by fungal adaptive strategies to the host microenvironment, including thermal dimorphism, cell wall remodeling associated with immune

evasion, and the capacity to persist within hostile tissue niches [8,9]. Subsequent evidence, largely derived from primary experimental studies and analyses of clinical isolates, expanded this framework by demonstrating that fungal persistence in PCM relies on coordinated adaptive responses to the host tissue microenvironment. Classical and contemporary cellular biology studies, predominantly conducted with *P. brasiliensis*, have established that metabolic reprogramming supports fungal survival under nutrient limitation [10], oxidative stress [11], and hypoxic conditions [12], while tightly regulated transcriptional responses linked to morphogenesis [13,14], stress adaptation [15], and long-term persistence contribute to chronic infection [16]. In parallel, structural and biochemical remodeling of the fungal cell surface modulates sensing and activation by host immune cells, promoting immune evasion and disease chronicity [17]. Building upon this knowledge, comparative investigations have shown that these adaptive programs are not uniform across the genus *Paracoccidioides*, with *P. brasiliensis* and *P. lutzii* exhibiting distinct metabolic profiles [18,19], cell wall compositions [20], and gene expression patterns [21,22] that differentially influence host interaction and immune recognition. In both species, these adaptive responses appear to converge on mechanisms that favor persistence within host tissues, albeit through species-specific regulatory and metabolic landscapes [23].

From a clinical perspective, infectious diseases, including fungal infections, are characterized by marked interindividual variability in susceptibility, clinical progression, and outcomes. Some Paracoccidioides species are clearly associated with distinct clinical manifestations [24], such as *P. ceti* and *P. lobogeorgii*, which cause chronic subcutaneous infections in dolphins [25] and humans [26], respectively. In contrast, in the context of human PCM, the available evidence does not support consistent differences in disease presentation associated with specific *Paracoccidioides* species, particularly within the *P. brasiliensis* species complex and *P. lutzii* [7,27] e *P. americana* [28]. Importantly, one major clinical implication of species diversity lies in its impact on diagnosis, particularly on serological assays [29]. Differences among *Paracoccidioides* species affect antigen expression profiles, such as the absence of gp43 production in *P. lutzii*, a major immunodominant antigen of *P. brasiliensis*, which may compromise the performance of conventional serological tests based on exoantigens derived from *P. brasiliensis* strains, potentially leading to false-negative results in patients infected with other species [30–33]. While fungal virulence factors and adaptive capacity are undoubtedly relevant to disease pathogenesis, growing evidence suggests that host heritable factors substantially contribute to interindividual variability in susceptibility, clinical presentation, and disease course [34–36].

The host immune response against *Paracoccidioides* spp. involves a coordinated interplay between innate and adaptive mechanisms to control and eliminate the infection. As reviewed by Burger [37], innate immunity provides the first line of defense, where neutrophils, macrophages, and natural killer (NK) cells recognize and respond to the fungus. Detection of fungal pathogen-associated molecular patterns (PAMPs) by pattern recognition receptors (PRRs), such as Toll-like receptors (TLRs) and Dectin 1, bridges innate and adaptive immunity by initiating inflammatory cascades and antigen presentation. The transition to adaptive immunity is critical for long-term control.

The control of *Paracoccidioides* spp. infection relies on an effective, antigen-specific cellular immune response, primarily mediated by T lymphocytes. In endemic areas, most infected individuals do not develop disease, exhibiting a robust Th1-type response characterized by cytokine production that activates macrophages and CD4+/CD8+T cells, leading to compact granuloma formation and fungal containment [38]. In contrast, patients who progress to severe forms (acute/subacute or severe disseminated chronic disease) display impaired Th1 immunity, with a predominant Th2/Th9 response, B-cell activation, hypergammaglobulinemia, and eosinophilia, compromising granuloma formation [39]. Quiescent fungal cells may persist within granulomas, potentially causing relapses.

Although susceptibility or resistance to infection is primarily mediated by the immune system, genetic factors can influence gene expression and downstream molecular pathways, thereby modulating how the host responds to infection and tissue damage [40]. Experimental murine models provide robust evidence that genetic control of resistance to *Paracoccidioides* infection is closely linked to the quality of the host's cellular immune response. In this context, resistant A/Sn mice display enhanced T cell activation, increased Interferon gamma (*IFNG*) production, and more efficient macrophage

function, in stark contrast to the immunological dysfunction observed in susceptible B10.A mice [41,42]. Genetic studies further demonstrated that this difference in susceptibility is associated with a dominant autosomal gene, mapped to the Pbc locus, which plays a critical role in regulating fungal burden and inflammatory responses [43,44]. This was confirmed by the observation that practically 100% of the first generation F1 hybrid had similar mortality data to those of the resistant parental mice.

The participation of the genetic pattern of the hosts was also shown by Xidieh et al. [45] who found strikingly different morphology of the granulomatous lesions in susceptible and resistant strains of mice, leading to contention of the fungi in the former and dissemination in the later groups. Moreover, the inheritance of resistance was shown to be a dominant trait, as mice from the F1 generation were resistant in terms of survival as well as in the development of the characteristic closed granulomas [44]. Finally, F1 hybrids presented persistent delayed type hypersensitivity reactions, known to be associated with control of paracoccidioidomycotic infection, similar to those observed in their resistant parental, at infection times when their susceptible parental was in a state of anergy.

These experimental findings have laid the foundation for investigating the role of host genetics in human PCM. Genetic susceptibility to infectious diseases is a key research area, as it may help elucidate mechanisms underlying disease progression and inform new therapeutic approaches. In the context of PCM, host genetic factors may influence the immune response to *Paracoccidioides* spp., determining the severity and progression of the disease [46]. Studies have suggested that genetic variants in immune-related genes, such as those encoding cytokines, pattern recognition receptors, and other immune modulators, may play a role in susceptibility to PCM [47–55].

This article is a systematic review of the existing literature on genetic susceptibility to PCM, with a specific focus on the association between single nucleotide variants (SNVs) and disease susceptibility, clinical manifestations, and severity.

## Methods

### Study design and data sources

We conducted a systematic review of literature published in scientific journals on March 13, 2024. Our study was guided by the Preferred Reporting Items for Systematic Reviews and Meta-Analyses (PRISMA) guidelines [56] (S1 PRISMA Checklist). We searched the following databases: MEDLINE (PubMed), Cochrane Library, LILACS (Latin American and Caribbean Health Sciences), SciELO, Web of Science, and Google Scholar. There were no language restrictions or date limits for the databases.

### Search strategy

To define the search strategy, we used synonyms of Health Sciences descriptors from the Health Sciences Descriptors (DeCS) (https://decs.bvsalud.org/) and the Boolean operators OR and AND. The keywords and concepts used were: "Blastomicose Sul-Americana" OR "Blastomicosis Sudamericana" OR "Blastomyces brasiliensis" OR "South OR American Blastomycosis" OR "Doença de Lutz-Splendore-Almeida" OR "Granuloma Paracoccidioide" OR "Granuloma Paracoccidioideo" OR "Granuloma Paracoccidioides" OR "Granulomatose Paracoccidióidica" OR "Paracoccidioides" OR "Paracoccidioidomicose" OR "Paracoccidioidomicosis" OR "Paracoccidioidomycosis" AND "Variante de Nucleotídeo Único" OR "Polimorfismo de Nucleotídeo Único" OR "Variante de Base Única" OR "Polimorfismo de Base Única" OR "Single Nucleotide Variant" OR "Single Nucleotide Polymorphism" OR "Variante de Nucleótido Simple" OR "Polimorfismo de Nucleótido Simple" OR "SNV" OR "SNVs" OR "SNP" OR "SNPs".

### Eligibility criteria, screening and data extraction

We included studies that investigated the association between SNVs and susceptibility to PCM. Additionally, we checked the reference lists of all included studies and relevant review articles identified through our search for additional

references. Two researchers (SSC and WSF) independently screened all search results and extracted data using a standard form. We strictly followed the recommendations of the Cochrane Handbook [57] and Joanna Briggs Institute [58] to assess risk of bias. For the case-control studies, we evaluated ten questions: (1) were the groups comparable other than the presence of disease in cases or the absence of disease in controls, (2) were cases and controls matched appropriately, (3) were the same criteria used for identification of cases and controls, (4) was exposure measured in a standard/valid and reliable way, (5) was exposure measured in the same way for cases and controls, (6) were confounding factors identified, (7) were strategies to deal with confounding factors stated, (8) were outcomes assessed in a standard/valid and reliable way for cases and controls, (9) was the exposure period of interest long enough to be meaningful, (10) and was appropriate statistical analysis used. We answered the questions using "Yes" "No", "Unclear", and or "Not Applicable". All authors independently conducted this assessment, and we resolved any disagreements by consensus between the reviewers.

We used the web-based Rayyan software for the initial screening of abstracts and titles [59]. We extracted information regarding SNV type, genotype, number of patients (cases), number of healthy individuals (controls), Odds Ratios (OR) values, the presence of SNV and PCM association, reporting of Hardy-Weinberg Equilibrium (HWE), genetic models tested, p-values correction, and covariates adjustment. We approached study authors to obtain additional data that were not available in the published articles, such as collection site and sample size. To ensure methodological consistency in both genetic association analyses and HWE evaluations, we used the SNPStats tool [60] to reanalyze all data. SNPStats enabled the evaluation of five distinct genetic models (dominant, recessive, additive, codominant, and overdominant), and through the Akaike Information Criterion (AIC) and the Bayesian Information Criterion (BIC), the software identified the best-fitting model for each variant. P-values were corrected using the False Discovery Rate (FDR) in R Studio software [61]. P-values < 0.05 were considered statistically significant.

We followed the HUGO Gene Nomenclature Committee guidelines for gene naming in this study [62].

## Meta-analysis

We performed a meta-analysis of studies utilizing identical SNVs (same locus and genotype). The Cochrane Handbook for Systematic Reviews of Interventions [57] dictates that, as a rule of thumb, tests for funnel plot asymmetry should be used only when there are at least 10 studies included in the meta-analysis, because when there are fewer studies the power of the tests is too low to distinguish chance from real asymmetry. In this way, we didn't carry out the funnel plot asymmetry test since only two studies analyzed the same SNVs. It was also not possible to adjust the analyses by age and sex because the authors did not indicate the genotypes separately for each individual. Heterogeneity across studies was assessed through Cochran's Q statistic and I² statistic. Using the Mantel-Haenszel method in the fixed-effect model, we obtained the overall or pooled Odds Ratio because it has better statistical properties when event rates are low and/or study sample size is small [57]. All analyses were carried out with IBM SPSS Statistics v 30.0 [63].

## Sample size calculation

The sample size calculation was performed considering a case-control design with a 1:1 ratio between cases and controls, a significance level of 5% ($\alpha = 0.05$), statistical power of 80% ($\beta = 0.20$), and a minimum relevant odds ratio (OR) of 2.0 [64]. The estimation was based on the classical formula for comparing two independent proportions [65]:

$$n = \frac{(Z_{\frac{\alpha}{2}} + Z_{\beta})^2 \, [p_1(1 - p_1) + p_2(1 - p_2)]}{(p_1 - p_2)^2}$$

(1)

where $n$ represents the number of individuals per group, $Z_{\alpha/2}$ is the critical value of the normal distribution for the significance level, $Z_{\beta}$ is the critical value for the desired power, $p_1$ is the expected proportion of exposure in the control group

(derived from the minor allele frequency in public database - https://www.ncbi.nlm.nih.gov/SNV/), and $p_2$ is the expected proportion in the case group, calculated from $p_1$ and the assumed OR using the following expression:

$$p_2 = \frac{OR \times p_1}{1 - p_1 + OR \times p_1}$$

(2)

The power analyses and verification of the required sample sizes were performed using the G*Power software (version 3.1.9.7, Universität Kiel, Germany), selecting the test for comparison of two independent proportions (test family: Exact; statistical test: Proportions: Inequality, two independent groups), with a two-tailed configuration and an allocation ratio of 1:1.

### Study Registration

International prospective register of systematic reviews (PROSPERO) (number CRD42025646417) (https://www.crd.york.ac.uk/PROSPERO).

## Results

### Genetic Associations of SNVs with PCM

The systematic search identified 710 records, with 666 (93.8%) sourced exclusively from PubMed, while other databases contributed minimally (Cochrane: 0, SciELO: 1, Web of Science: 10). After removing 24 duplicates, 686 unique records were screened. 677 records (98.7%) were excluded during screening, leaving only 9 articles for full-text assessment. Of these, 2 were excluded for not addressing the association between SNVs and PCM susceptibility. Ultimately, 7 studies were included in the qualitative synthesis (Fig 1).

The studies examined the role of 11 genes and 15 SNVs in the disease's development (Table 1). These studies analyzed an average of 92 samples from patients with PCM (ranging from 49 to 156) and 87 samples from healthy individuals (ranging from 31 to 121) who did not develop the disease. Only two studies included individuals in the control group who had prior exposure to *Paracoccidioides* sp. These individuals were identified through lymphoproliferation assays against the 43-kDa glycoprotein of *P. brasiliensis* [52] or skin tests using paracoccidioidin and histoplasmin [53].

All the studies were conducted in Brazil, specifically in the central-west and southeast regions, which are the high endemic areas for the disease, with the states of São Paulo and Minas Gerais having the largest sample sizes and the highest number of analyzed SNVs (Fig 2). Two studies did not report the origin and number of samples in each group (case and control) [52,53]. After contacting the authors, we were able to retrieve this data only from Lozano et al. [52] and the data are presented in S3 Table. The risk of bias assessment revealed variable methodological quality among the analyzed studies, with most presenting high or moderate risk of bias (Fig 3).

These studies selected key genes involved in immune response to *Paracoccidioides* spp., such as fungal receptors like CD209 molecule (*CD209*), and Fc gamma receptor IIa (*FCGR2A*), pro-inflammatory cytokines and their signaling molecules, e.g., Tumor Necrosis Factor (*TNF*), Interleukin 18 (*IL18*), *IFNG*, Interleukin (*IL12*), Janus Kinase 1 (*JAK1*), and anti-inflammatory or regulatory factors like Interleukin 10 (*IL10*), Cytotoxic T-Lymphocyte Associated protein 4 (*CTLA4*), and Vitamin D Receptor (*VDR*). The SNVs are localized in the 5' up-stream regions of cytokines genes (*IL10, TNF, IL4, IL12A, IFNG, IL18*), receptors (*FCGR2A, CD209*), and regulatory or signaling molecules (*CTLA4, JAK1*), in the 3' UTR (*IL12B*) or in the intronic sequence (*IFNG, VDR*). Just one polymorphism is a missense variant in the coding region of the *CTLA4* gene (Table 1).

The study of Bozzi et al. [48] aimed to investigate the frequency of genotypes with the Tumor Necrosis Factor α (*TNFα* rs1800629) gene variant G/A at position -308 and the Interleukin 10 (*IL10* rs1800896) gene variant G/A at position -1082, and to verify a possible association of these variants with PCM. The authors suggested that GG genotype at position -1082 of *IL10* gene could be a risk factor to an increased susceptibility to PCM.

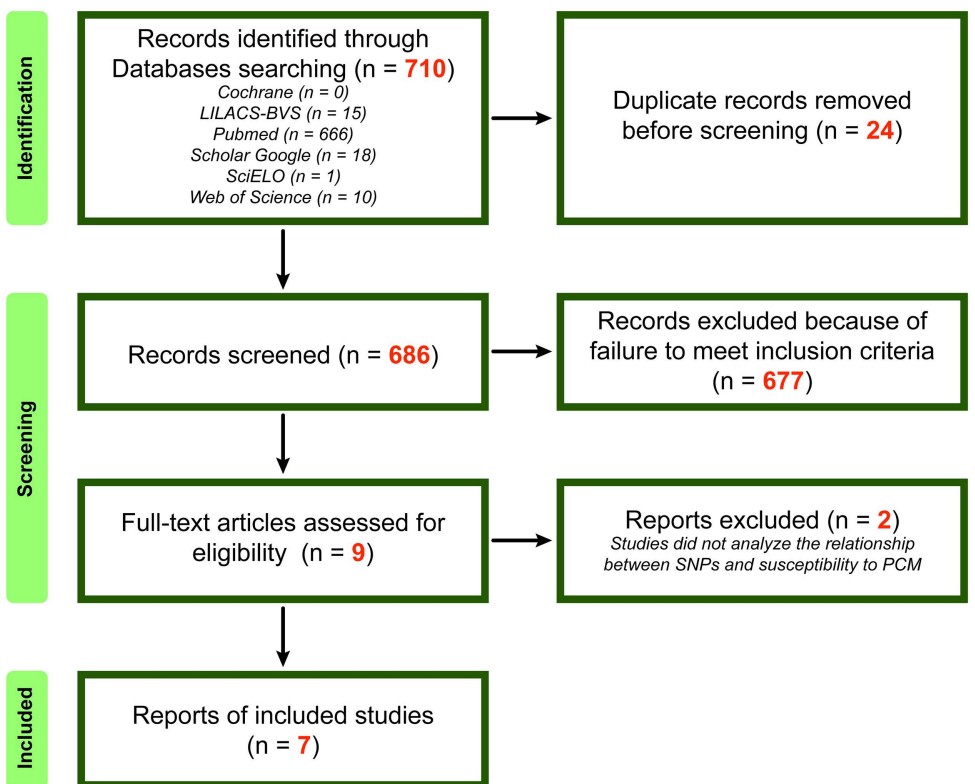

**Fig 1. PRISMA (Preferred Reporting Items for Systematic Reviews and Meta-Analyses) flow diagram of the studies identified, screened, and assessed for eligibility.**

Another study by the same research group [49], investigated the genetic variation in the Interferon gamma (*IFNG* rs2430561) and Interleukin 4 (*IL4* rs2243250) genes, which encode cytokines related to Th1 and Th2 immune responses, respectively, and evaluated their influence on susceptibility to PCM. The SNV of -590 *IL4* showed that C/T genotype was significantly more prevalent in PCM group. Although the data suggested a possible association between genetic variants and the disease, the results did not indicate statistically significant differences in the SNVs analyzed, pointing to the need for further investigations.

Lozano et al. [52] investigated the possible association between the SNVs in Cytotoxic T-Lymphocyte Associated protein 4 (*CTLA4*) gene and the risk of developing PCM. Their results did not indicate any significant association between the two SNVs analyzed (318C/T rs2243250 in the promoter and +49A/G rs5742909 in exon 1) and the occurrence of PCM.

Mendonça et al. [53] investigating two variants in the *IL4* gene rs2243250 (-590 C/T and microsatellite in intron-3), highlighted that there was a significant correlation between the RP2/RP2 intron-3 genotype and infection with *Paracoccidioides* sp., whereas the RP1/RP1 genotype was correlated with resistance. However, no significant correlation was observed for the *IL4* promoter variant. Furthermore, the authors did not specify the origin and quantity of samples for each group. They mention that the study was conducted at two Brazilian centers: the General Hospital of the Federal University of Triângulo Mineiro, Minas Gerais State and the University Hospital of Botucatu Medical School, São Paulo State University (S3 Table).

Carvalho et al. [50] conducted a study to investigate the incidence of variants in the Interleukin 12B (*IL12B* rs3212227), *IL12RB1* rs11575834, and *IFNG* rs2430561 genes in different clinical forms of PCM and found no genetic association.

**Table 1. Summary of the results of the seven studies eligible for the systematic review on genetic susceptibility to Paracoccidioidomycosis (PCM).** The data shows the name of the gene, the mutation position, the reference SVP ID, the genotype, the number (N) and percentage (%) of patients with PCM and healthy individuals (control), Hardy-Weinberg Equilibrium (HWE) Deviation, Genetic Model tested, the Odds Ratio, 95% Confidence Interval (95% CI), p-value, p-value adjustment type, Minor Allele Frequence (MAF), observed (OSS) and expected sample size (ESS), and reference. NS: not significant. NR: Not Reported. CTLA4: cytotoxic T-lymphocyte associated protein 4; CD209: CD209 molecule; FCGR2A: Fc gamma receptor IIa; IFNG: interferon gamma; IL10: interleukin 10; IL12A: interleukin 12A; IL12B: interleukin 12B; IL12RB1: interleukin 12 receptor subunit beta 1; IL18: interleukin 18; IL4: interleukin 4; JAK1: Janus kinase 1; TNF: tumor necrosis factor; VDR: vitamin D receptor. ¹HWE Deviation: *No deviation.

| Gene Reference SNV ID Mutation at position [Article Reference] | Genotypes (HWE Deviation¹) | Patients with PCM N (%) | Healthy individuals N (%) | Odds Ratio [95% CI] p-value/ p-value adjustment | Genetic Model | MAF Global (OSS/ESS) |
|---|---|---|---|---|---|---|
| IL10 rs1800896 -1082G>A [48] | GG (NR) | 14(29) | 2(6) | 5.8 [NR] NR/ NR | NR | 0.452812 (80/320) |
| | GA (NR) | 25(51) | 17(55) | NR [NR] NR/ NR | | |
| | AA (NR) | 10(20) | 12(39) | | | |
| TNF rs1800629 -308G>A [48] | GG (NR) | 46(85) | 21(68) | NR [NR] NR/ NR | NR | 0.15204 (85/486) |
| | GA (NR) | 6(11) | 9(29) | | | |
| | AA (NR) | 2(4) | 1(3) | | | |
| IFNG rs2430561 +874T>A [48] | AA (NR) | 13(26) | 11(36) | NR [NR] NR/ NR | NR | 0.39991 (81/320) |
| | TA (NR) | 28(56) | 14(45) | | | |
| | TT (NR) | 9(18) | 6(19) | | | |
| IL4 rs2243250 -590C>T [48] | TT (NR) | 1(2) | 0(0) | NR [NR] NR/ NR | NR | 0.185405 (81/428) |
| | CT (NR) | 20(39) | 6(19) | | | |
| | CC (NR) | 29(59) | 25(81) | | | |
| CTLA4 rs5742909 -318C/T [52] | CC (*) | 57(86) | 72(95) | 0.35 [0.10-1.20] 0.08/ NR | NR | 0.082449 (142/788) |
| | CT (*) | 9(14) | 4(5) | 2.84 [0.83-9.71] 0.08/ NR | | |
| | TT (*) | 0(0) | 0(0) | NR [NR] NR/ NR | | |
| CTLA4 rs231775 +49A/G [52] | AA (*) | 29(44) | 36(47) | 0.87 [0.45-1.69] 0.68/ NR | NR | 0.372277 (142/320) |
| | AG (*) | 29(44) | 34(45) | 0.97 [0.50-1.88] 0.92/ NR | | |
| | GG (*) | 8(12) | 6(8) | 1.61 [0.53-4.91] 0.40/ NR | | |
| IL4 rs2243250 -590C/T [53] | CC (NR) | 8(40.52) | 15(20.54) | NR [NR] NR/ NR | NR | 0.185405 (149/428) |
| | CT (NR) | 22(28.95) | 21(28.76) | | | |
| | TT (NR) | 46(60.53) | 37(50.68) | | | |
| IFNG rs2430561 +874T>A [50] | AA (*) | 66(42.3) | 48(39.21) | NR [NR] NR/ NR | NR | 0.39991 (277/320) |
| | TA (*) | 69(44.2) | 57(47.1) | | | |
| | TT (*) | 21(13.5) | 16(13.2) | | | |
| IL12B rs3212227 +1188A/C [50] | AA (*) | 82(52.6) | 55(45.4) | NR [NR] NR/ NR | NR | 0.21535 (277/394) |
| | AC (*) | 59(37.8) | 52(43) | | | |
| | CC (*) | 15(9.6) | 14(11.6) | | | |
| IL12RB rs11575834 641A>G [50] | AA (*) | 95(60.9) | 75(62) | NR [NR] NR/ NR | NR | 0.19408 (277/416) |
| | AG (*) | 53(34) | 35(28.9) | | | |
| | GG (*) | 8(5.1) | 11(9.1) | | | |

*(Continued)*

**Table 1.** (Continued)

| Gene<br>Reference SNV ID<br>Mutation at position<br>[Article Reference] | Genotypes<br>(HWE Deviation¹) | Patients with<br>PCM N (%) | Healthy<br>individuals<br>N (%) | Odds Ratio [95% CI]<br>p-value/ p-value<br>adjustment | Genetic<br>Model | MAF Global<br>(OSS/ESS) |
|---|---|---|---|---|---|---|
| TNF<br>rs1800629<br>-308G>A<br>[47] | AA (*) | 1(1.75) | 2(2.33) | 0.73 [0.04-12.32]<br>0.804/ NR | Codominant | 0.15204<br>(143/486) |
| | AG (*) | 6(10.53) | 12(13.95) | 0.54 [0.16-1.82]<br>0.804/ NR | | |
| | GG (*) | 50(87.72) | 72(83.72) | 1.00 [NR]<br>0.804/ NR | | |
| | GG vs AA+AG (*) | 7(12.29) | 14(16.28) | 0.57 [0.18-1.76]<br>0.632/ NR | Dominant | |
| | GG+AG vs AA (*) | 56(98.25) | 84(97.68) | 0.80 [0.05-13.43]<br>NS/ NR | Recessive | |
| JAK1<br>rs11208534<br>-301A>G<br>[47] | AA (*) | 41(66.13) | 71(68.27) | 1.00 [NR]<br>0.377/ NR | Codominant | 0.126297<br>(166/558) |
| | AG (*) | 19(30.65) | 25(24.04) | 1.51 [0.64-3.55]<br>0.377/ NR | | |
| | GG (*) | 2(3.22) | 8(7.69) | 0.69 [0.15-3.12]<br>0.377/ NR | | |
| | AA vs GG+AG (*) | 21(33.87) | 33(31.73) | 1.27 [0.57-2.77]<br>0.864/ NR | Dominant | |
| | AA+AG vs GG (*) | 60(96.78) | 96(92.31) | 0.63 [0.14-2.78]<br>0.324/ NR | Recessive | |
| FCGR2A<br>rs1801274<br>-519G>A [47] | AA (*) | 11(21.15) | 18(17.65) | 1.54 [0.50-4.70]<br>0.747/ NR | Codominant | 0.489927<br>(153/324) |
| | AG (*) | 20(39.22) | 46(45.10) | 0.93 [0.40-2.17]<br>0.747/ NR | | |
| | GG (*) | 20(39.23) | 38(37.25) | 1.00 [NR]<br>0.747/ NR | | |
| | GG vs AA+AG (*) | 31(60.78) | 64(62.75) | 1.07 [0.49-2.35]<br>0.861/ NR | Dominant | |
| | GG+AG vs AA (*) | 41(78.84) | 84(82.35) | 1.60 [0.58-4.43]<br>0.665/ NR | Recessive | |
| CD209<br>rs4804803<br>-336A>G [47] | AA (*) | 32(64.00) | 70(67.31) | 1.00 [NR]<br>0.032/ Bonferroni | Codominant | 0.213316<br>(154/390) |
| | AG (*) | 8(16.00) | 27(25.96) | 0.26 [0.28-2.04]<br>0.032/ | | |
| | GG (*) | 10(20.00) | 7(6.73) | 3.76 [1.06-13.38]<br>0.032/ | | |
| | AA vs GG+AG (*) | 18(36.00) | 34(32.69) | 1.33 [0.59-3.04]<br>0.718/ NR | Dominant | |
| | AA+AG vs GG (*) | 40(80.00) | 97(93.27) | 4.04 [1.16-14.13]<br>0.025/ Bonferroni | Recessive | |
| VDR<br>rs7975232<br>+64978A>C [47] | AA (*) | 12(22.22) | 33(39.76) | 1.00 [NR]<br><0.001/ Bonferroni | Codominant | 0.4455205<br>(137/320) |
| | AC (*) | 9(16.67) | 36(43.37) | 0.86 [0.27-2.74]<br><0.001/ | | |
| | CC (*) | 33(61.11) | 14(16.87) | 5.94 [2.07-17.05]<br><0.001/ | | |
| | AA vs CC+AC (*) | 42(77.78) | 50(60.24) | 2.71 [1.07-6.86]<br>0.041/ Bonferroni | Dominant | |
| | AA+AC vs CC (*) | 21(28.89) | 69(83.13) | 6.36 [2.52-15.94]<br><0.001/ Bonferroni | Recessive | |

*(Continued)*

Diseases

**Table 1.** (Continued)

| Gene<br>Reference SNV ID<br>Mutation at position<br>[Article Reference] | Genotypes<br>(HWE Deviation¹) | Patients with<br>PCM N (%) | Healthy<br>individuals<br>N (%) | Odds Ratio [95% CI]<br>*p-value/ p-value*<br>*adjustment* | Genetic<br>Model | MAF Global<br>(OSS/ESS) |
|---|---|---|---|---|---|---|
| *IL12A*<br>rs2243115<br>-504G>T<br>[55] | GG (*) | 2(1.3) | 2(1.8) | 1.00 [-]<br>NS/ NR | Codominant | 0.112035<br>(259/614) |
| | GT (*) | 40(26.9) | 28(25.5) | 1.43 [0.19-10.76]<br>NS/ NR | | |
| | TT (*) | 107(71.8) | 80(72.7) | 1.34 [0.18-9.7]<br>NS/ NR | | |
| *IL18*<br>rs1946518<br>-607C>A<br>[55] | CC (*) | 44(29.5) | 29(26.4) | 1.00 [-]<br>NS/ NR | Codominant | 0.408969<br>(259/318) |
| | CA (*) | 68(45.6) | 60(54.5) | 0.75 [0.42-1.34]<br>NS/ NR | | |
| | AA (*) | 37(24.8) | 21(19.1) | 1.16 [0.57-2.37]<br>NS/ NR | | |
| *IFNG*<br>rs1327474<br>-611A>G<br>[55] | AA (*) | 21(14.1) | 14(12.8) | 1.00 [-]<br>NS/ NR | Codominant | 0.425059<br>(258/318) |
| | AG (*) | 66(44.3) | 47(43.1) | 0.93 [0.43-2.03]<br>NS/ NR | | |
| | GG (*) | 62(41.6) | 48(44.0) | 0.86 [0.40-1.87]<br>NS/ NR | | |

The study of Alves Pereira Neto et al. [47] aimed to elucidate how the genetic variations might influence susceptibility to chronic PCM with oral manifestations. The investigation specifically focused on SNVs in *CD209* molecule (*CD209* rs4804803) and Vitamin D Receptor (*VDR* rs7975232) genes, which play established roles in fungal immune responses. Notably, the study identified that the CC genotype of *VDR* and the GG genotype of *CD209* were associated with increased risk of oral PCM. These findings highlighted these SNVs as potential biomarkers for identifying high-risk individuals. However, after reanalyzing the data, we identified that there was a significant association only between the *VDR* SNV and susceptibility against PCM (S1 Table).

In the last work included in this review, Sato et al. [55] investigated the variant in the Interleukin 2p35 (*IL12A* rs2243115), Interleukin 18 (*IL18* rs1946518), and *IFNG* rs1327474 genes and the association with severity on PCM but found no significant association.

When we reanalyzed the genotypes distribution of the 15 SNPs, we observed that three of them, namely *IL4* rs2243250 [49]; *IL12* receptor subunit beta 1 (*IL12RB1*) rs11575834 [50], and *JAK1* rs11208534 [47] showed a significant deviation from the HWE in the control group, but not in patients (S1 Table).

Regarding genetic association, only two SNVs showed a significant association between the patients' genotype and their susceptibility or resistance to developing PCM. The data suggests that variant in the *IL10* gene (rs1800896) could be a risk factor for increased susceptibility to PCM (OR = 2.52, 95% CI = 1.22-5.23, $p_{FDR}$ = 0.025, AIC/BIC = 104.0/108.7) [48]. On the other hand, SNV in the *VDR* gene (rs7975232) may be related to protection in the development of PCM (OR = 0.13, 95% CI = 0.06-0.29, $p_{FDR}$ = 0.00017, AIC/BIC = 159.0/164.9) [47]. However, these studies did not evaluate the association between genetic variants and the clinical form or severity of PCM, nor did they assess pathogen exposure using standardized, valid, and reliable methods, such as intradermal testing with paracoccidioidin or gp43. Consequently, the interpretation of their conclusions should be approached with caution, as potential confounding factors were not adequately identified or controlled. The results of the genetic association analyses and Hardy-Weinberg equilibrium tests for the remaining SNVs are provided in the supplementary material (S1 Table).

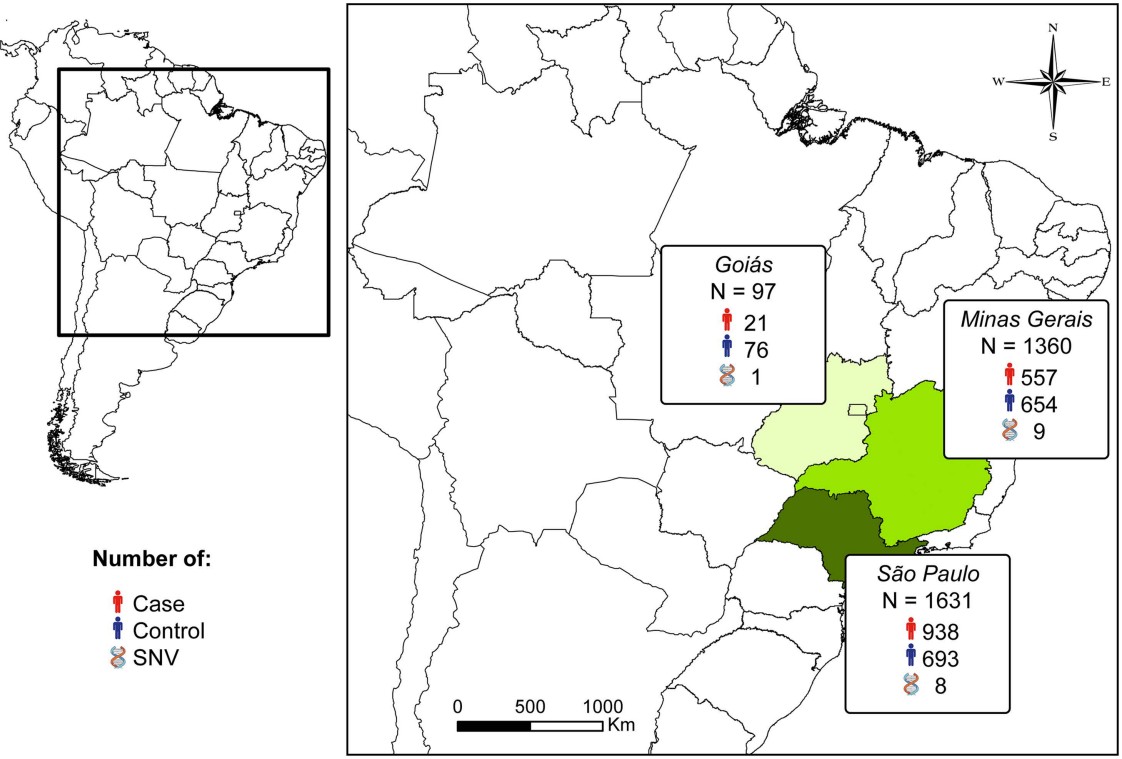

**Fig 2. Sample size and collection site of case and control groups, as well as the number of Single Nucleotide Variants analyzed in seven studies that evaluated genetic susceptibility to Paracoccidioidomycosis in Brazil.** The map was generated using ArcGIS 10.1 (ESRI, California) and GADM layers (https://gadm.org/download_country.html) under CC BY 4.0 license. Icons were were taken from the World Atlas of Wikimedia (https://commons.wikimedia.org; **DNA:** https://commons.wikimedia.org/wiki/File:202202_DNA_colored.svg; **Man:** https://commons.wikimedia.org/wiki/File:-Frankie_Pappas_Icon.png**).**

## Meta-analysis

Results of meta-analysis are shown in Figs 4-6. We identified three SNVs that were evaluated in at least two studies: *IFNG* [49,50] (Fig 4), *IL4* [48,52] (Fig 5), and *TNFα* [47,48] (Fig 6). We found significant associations only in the *IL4* SNV with PCM resistance for genotype CC, as demonstrated by the forest plot analysis (Fig 5). However, upon reanalyzing the data used in the meta-analysis in SNPStats, we found no significant association between any SNV and the development of PCM. Furthermore, the population studied for the *IL4* SNV was not in HWE (S2 Table).

## Discussion

Our systematic review evaluated the association between single nucleotide variants and susceptibility to PCM across seven studies. After reanalyzing the data from the retrieved articles, we found a statistically significant association between specific SNVs only in two key immune-related genes: Interleukin 10 and Vitamin D Receptor. Although the meta-analysis (forest plot) identified a significant association between *IL4* rs2243250 variant and PCM, this result cannot be considered valid because the population is not in Hardy-Weinberg equilibrium.

The significant associations identified in our review highlight the importance of understanding the genetic variability that influences immune responses in PCM. The *VDR* gene plays a pivotal role in immune response modulation and susceptibility to infectious diseases, including tuberculosis and HIV [55]. *VDR* is essential for mediating the immuno-modulatory effects of vitamin D, which enhances antimicrobial defense mechanisms. Notably, vitamin D upregulates

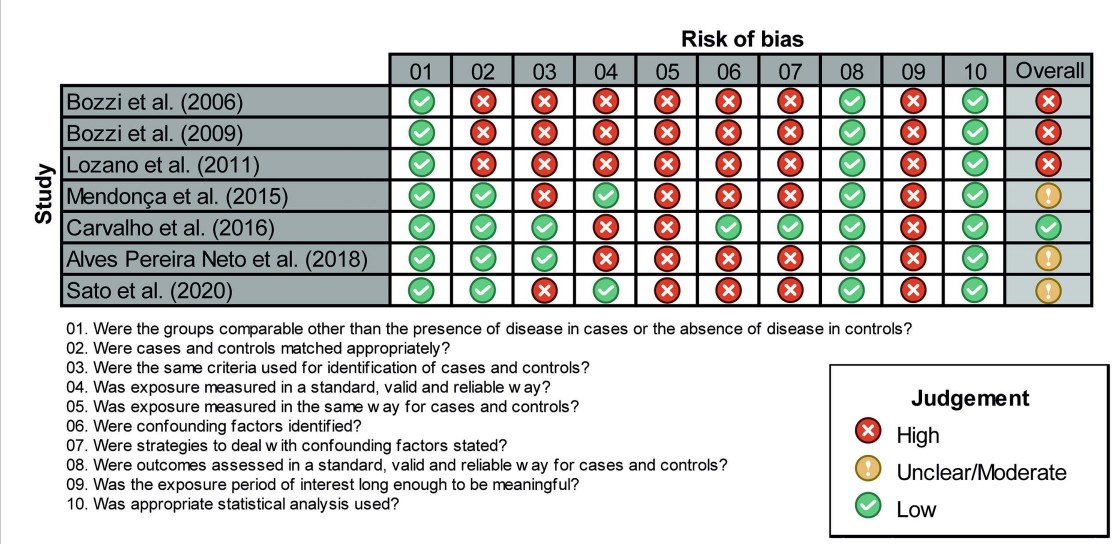

**Fig 3. Risk of bias assessment of seven case-control studies included in the systematic review, based on the Joanna Briggs Institute (JBI) Critical Appraisal Checklist.** Most studies were classified as presenting moderate to high risk of bias.

the production of antimicrobial peptides (e.g., cathelicidin), thereby promoting chemotaxis and phagocytic activity in immune cells such as macrophages and neutrophils [66,67]. Genetic variation in the VDR gene has been consistently associated with differences in vitamin D signaling and susceptibility to disease. The polymorphisms most frequently investigated, e.g., *FokI* (rs2228570), *TaqI* (rs731236), *ApaI* (rs7975232), and *BsmI* (rs1544410), affect distinct regulatory and functional aspects of VDR expression and activity [68]. Specifically, rs2228570, located in exon 2, and rs731236, located in exon 9, may influence translation efficiency and VDR protein structure, whereas rs7975232 and rs1544410, both located in intron 8, have been associated with altered mRNA stability and reduced transcriptional efficiency [69]. Together, these variants may lead to decreased VDR activity and impaired vitamin D function, providing a plausible biological basis for their reported associations with increased susceptibility to infectious and immune-mediated diseases [70]. In the context of PCM, the role of vitamin D remains poorly understood [71]. Nevertheless, cases of hypercalcemia have been reported in disseminated PCM, a condition that has been linked to the production of the biologically active form of vitamin D (1,25 hydroxyvitamin D) [72,73]. Alves Pereira Neto et al. [47] reported a significant association between the *Apa1* variant (rs7975232) and increased PCM susceptibility, suggesting a potential role for *VDR* variants in host-pathogen interactions. They further explored correlations between SNVs and granuloma organization in oral lesions, providing insights into disease severity and progression. However, the authors do not discuss the functional impacts of these genetic variants on the disease.

Interleukin 10 is an immunosuppressive and anti-inflammatory cytokine, essential in regulating the immune response, primarily acting on the modulation of pro-inflammatory cytokines, such as those produced by Th1 cells. In PCM, *IL10* exerts an immunosuppressive effect that can influence both resistance and susceptibility to infection [38]. Its excessive production can limit the effective immune response against the fungus, facilitating the persistence of the pathogen, while controlled levels can prevent tissue damage [74]. The *IL10* gene is highly polymorphic, with multiple promoter-region variants (including -592A/C, -819C/T, and -1082G/A) known to modulate transcription levels and influence susceptibility to infectious diseases like tuberculosis and leprosy [75,76]. Genetic variation in *IL10* has also been found to be the underlying cause of susceptibility toward fungal infections like Invasive Aspergillosis [77], *Candida* and *Coccidioides immitis* infection [78].

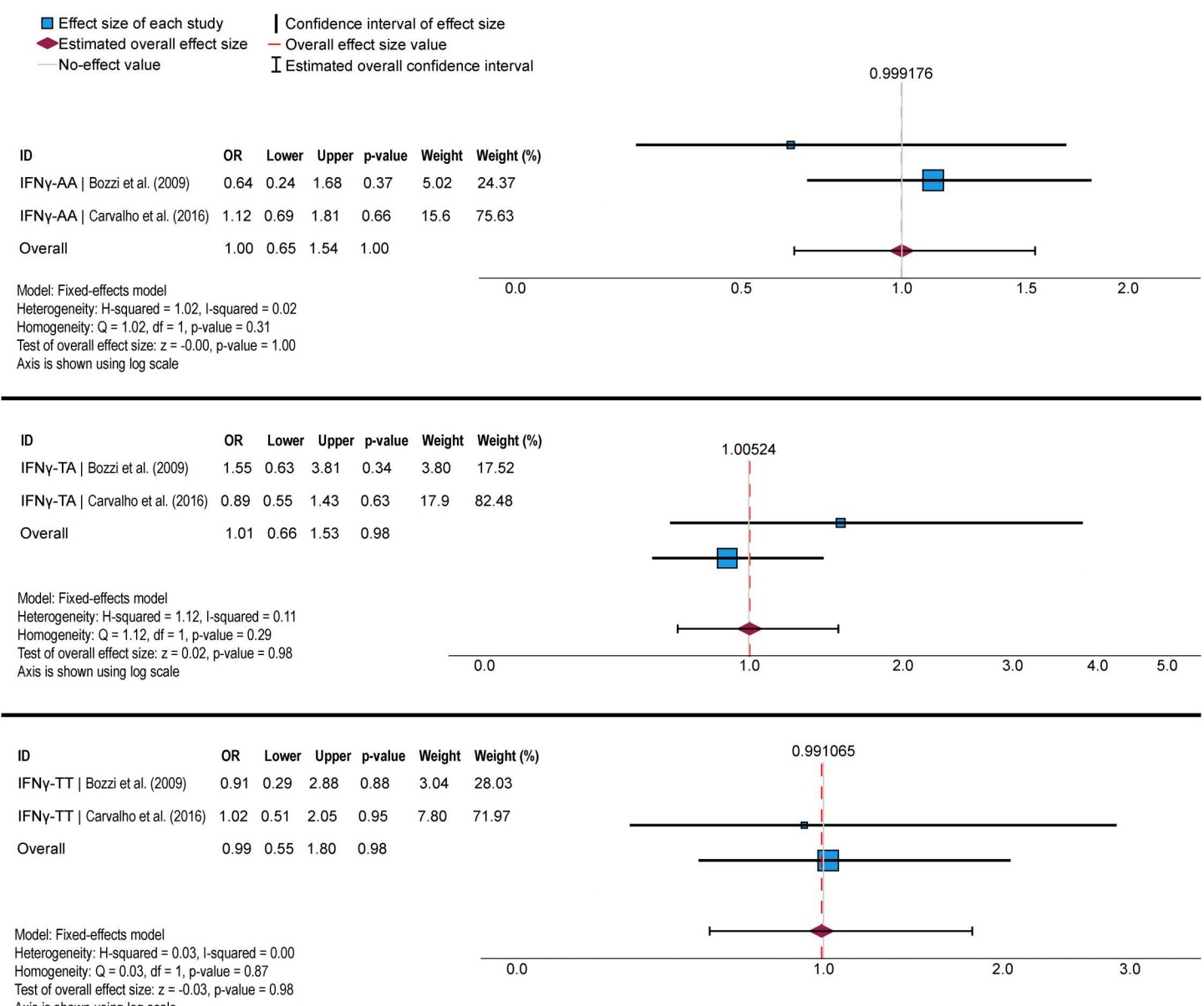

**Fig 4. Meta-analysis of two case-control genetic association studies evaluating the Interferon gamma (IFNG rs2430561) single nucleotide variant (SNV) and susceptibility to paracoccidioidomycosis, considering the AA, TA, and TT genotypes.** Meta-analysis was performed using the Mantel-Haenszel method in the fixed-effect model. The size of each box indicates the weight of the study in the pooled results. OR: odds ratio.

While the −592A/C and −819C/T SNVs have been significantly associated with leprosy susceptibility [76], Bozzi et al. [48] identified the −1082G/G (rs1800896) genotype as a potential risk factor for PCM. However, the absence of reported 95% confidence intervals in their study limits the assessment of the precision and magnitude of this association [79]. The authors also discussed the possible influence of *IL-10* SNVs on cytokine production but did not reach a definitive conclusion regarding their functional impact. Notably, independent evidence indicates that the G allele of rs1800896 (−1082/−1117) can modulate *IL10* production in severe cases of bronchiolitis caused by respiratory

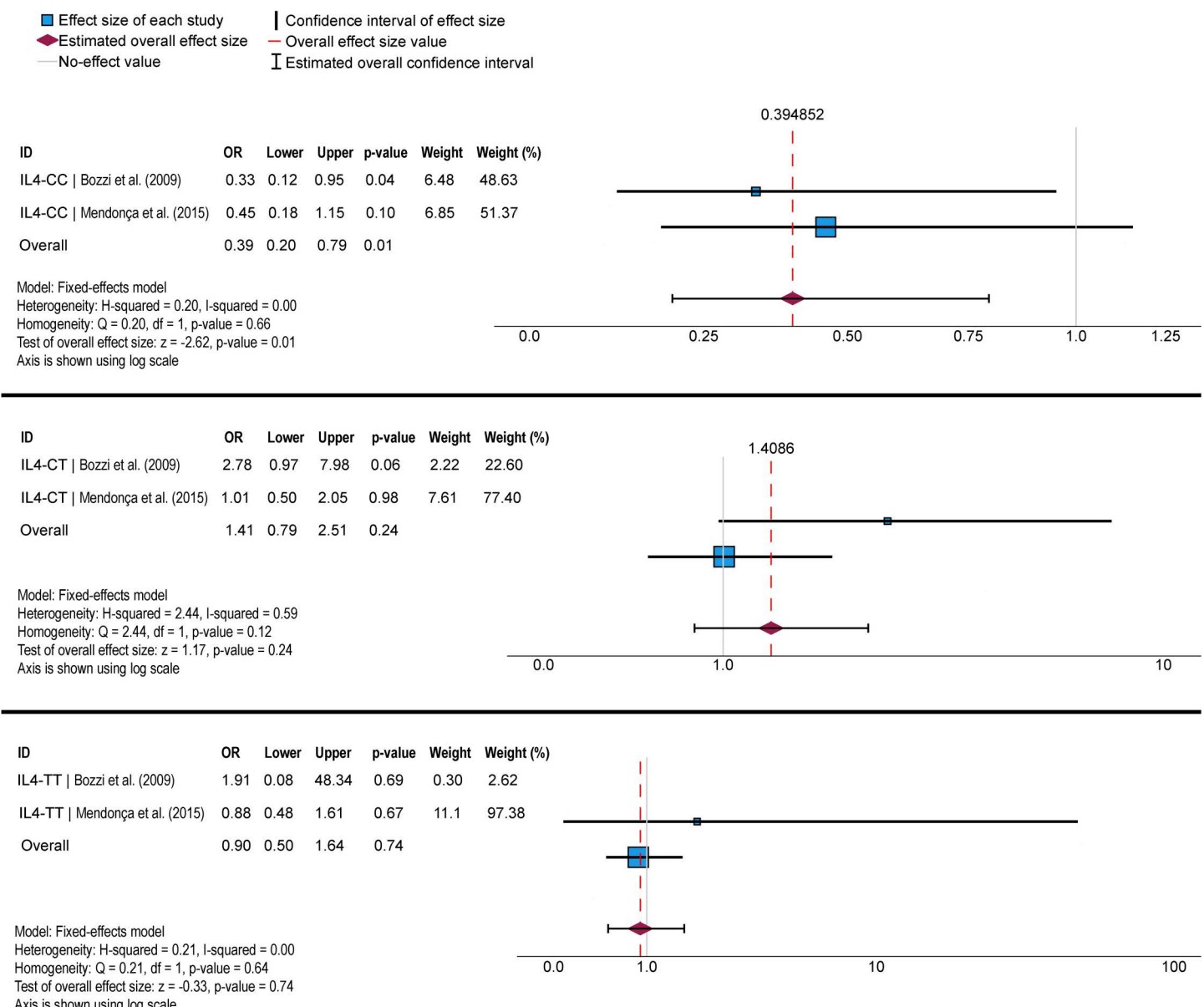

**Fig 5. Meta-analysis of two case-control genetic association studies evaluating the Interleukin 4 (IL4 rs2243250) single nucleotide variant (SNV) and susceptibility to paracoccidioidomycosis, considering the CC, CT, and TT genotypes.** Meta-analysis was performed using the Mantel-Haenszel method in the fixed-effect model. The size of each box indicates the weight of the study in the pooled results. OR: odds ratio.

syncytial virus, leading to an inadequate anti-inflammatory response and potentially contributing to uncontrolled inflammation and more severe respiratory symptoms [80]. Although indirect, these observations provide biological support for a possible role of rs1800896 in disease susceptibility and severity through altered *IL10*-mediated immune regulation.

An important limitation of the meta-analyses conducted in this review is the inclusion of only two studies per SNV, leading to inherently low statistical power [79] and preventing reliable assessment of publication bias through funnel plot

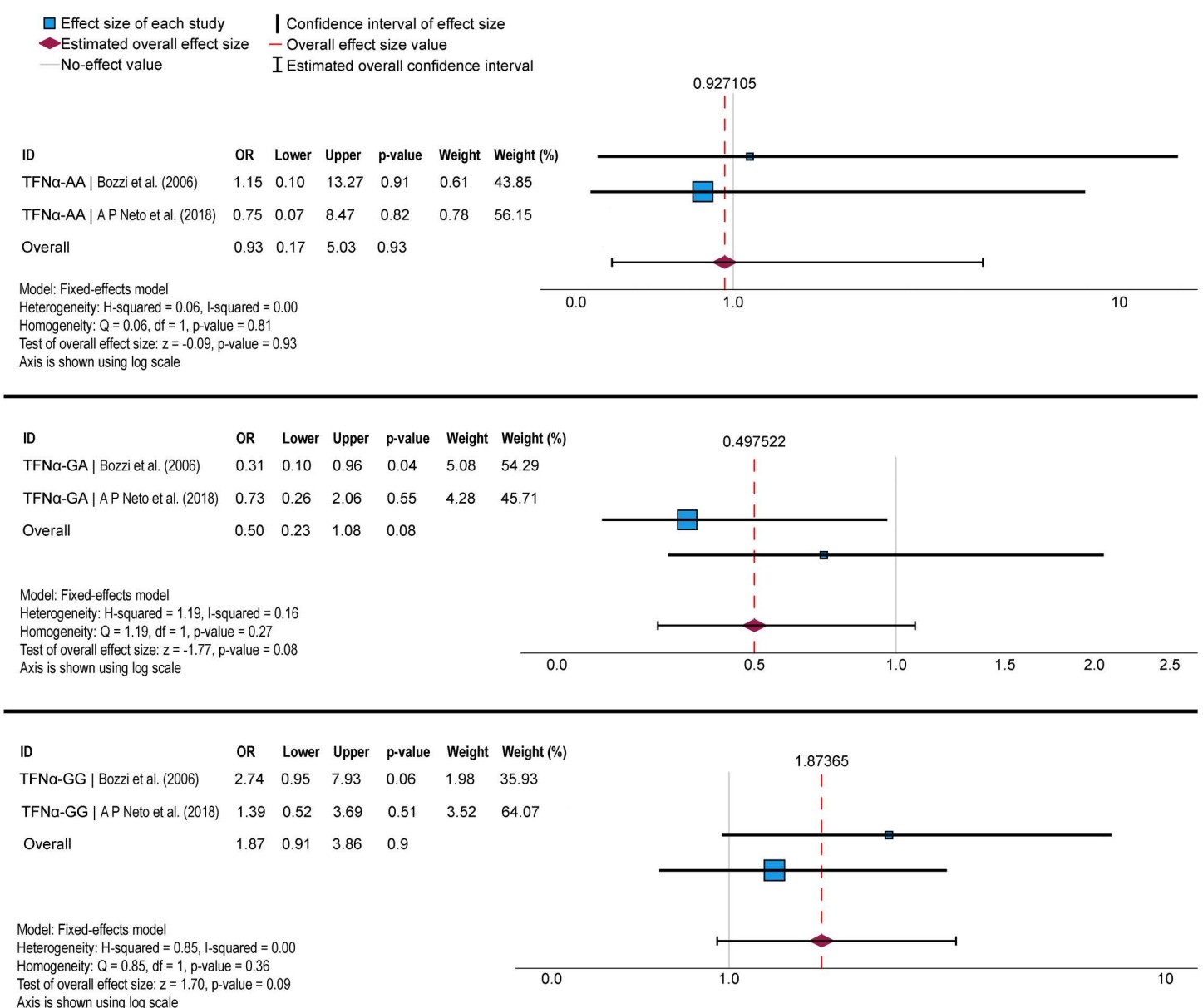

**Fig 6. Meta-analysis of two case-control genetic association studies evaluating the Tumor Necrosis Factorα (TNFα rs1800629) single nucleotide variant (SNV) and susceptibility to paracoccidioidomycosis, considering the AA, GA, and GG genotypes.** Meta-analysis was performed using the Mantel-Haenszel method in the fixed-effect model. The size of each box indicates the weight of the study in the pooled results. OR: odds ratio.

asymmetry tests, which are not informative when few studies are available [49]. Consequently, the absence of significant associations in the pooled analyses should be interpreted with caution and does not exclude a potential biological role of these variants in PCM susceptibility. Rather, this finding reflects the scarcity of replicated genetic association studies in PCM and underscores the fragmented nature of the current evidence. In this context, the meta-analysis serves an exploratory purpose, helping to prevent overinterpretation of isolated positive findings and highlighting critical gaps in the

literature that must be addressed by future, adequately powered studies with standardized designs and well-characterized exposure-based control groups.

## Methodological limitations of the reviewed studies

**Genetic background.** The primary challenge in case-control studies is ensuring that the genetic backgrounds of cases and controls are well-matched, so any genetic differences are related to the disease and not due to biased sampling. Therefore, cases and controls should ideally belong to similar ethnic groups. To avoid more subtle genetic differences, controls can be selected from the same geographical area as the cases, or information such as the birthplace of grandparents can be used to ensure a similar distribution between cases and controls [80].

Regarding ethnic groups, three studies discussed their implication in the association between SNVs and PCM. Lozano et al. [52] estimated the individual ancestry in the patient and control groups by Bayesian inference and found no significant difference, confirming the admixture-matched contribution of ancestries for the two groups. Carvalho et al. [50] found a significant difference between the distributions of genotypic/allelic frequencies and ethnic groups. Nevertheless, the authors discussed that further studies are necessary due to the small number of individuals herein. Finally, Sato et al. [55] found no differences on allelic and genotypic distributions between patients and controls or between acute form and chronic form groups of patients and showed that the possible limitations in the study were the inclusion of only one health center, mixed ethnic groups in the Brazilian population, and lack of detection of *IL18* levels and functional analyses.

**Sample size.** Another limitation found in the studies concerns the sample size analyzed. Lozano et al. [52], Mendonça et al. [53], and Carvalho et al. [50] state that the lack of a significant association between genotypes and the development of PCM may be related to the reduced sample size. However, in hypothesis tests, small-sample studies can provide better evidence for treatment than large-sample studies, given that equal P-values have been obtained [81,82].

Although none of the seven studies selected in our review calculated the minimum required sample size, using Schlesselman's formula [65] we demonstrated that none of the articles achieved the minimum sample size necessary to find a statistically significant association (see Table 1). However, even with insufficient sampling, Bozzi et al. [48] and Alves Pereira Neto et al. [47] found significant genetic associations between PCM and variations in the *IL10* and *VDR* SNVs, respectively. Sample size, and consequently the precision of the estimates, have a strong influence on how research results are interpreted [83]. Therefore, instead of treating sample size merely as a way to achieve statistical significance, researchers should interpret their results in light of estimate precision, acknowledging uncertainty and considering the full range of plausible effects, regardless of whether the P-value crosses a conventional threshold.

**Validity and robustness of the statistical analysis.** The Hardy-Weinberg Equilibrium (HWE) is a fundamental principle in population genetics, describing the expected genotypic frequencies in a population under specific conditions. Disease-free control groups from outbred populations are expected to conform to HWE. Deviations from HWE in control groups may indicate problems such as genotyping errors, population stratification, or selection bias [84].

In our review, after reanalyzing the data, we found that the control groups for the *IL4* [53], *IL12RB1* [50], and *JAK1* [47] SNVs were not in HWE. Additionally, the case group in studies of the association between PCM and SNVs in the *CD209* and *VDR* genes [47] also deviated from HWE (S1 Table). Although HWE testing is a useful analytical tool for identifying potential issues in genetic association studies, a group not in HWE may still be used if the reasons for the deviation are understood and accounted for. This is especially true for disease groups where the disease itself may cause deviations, or if the deviation is due to correctable issues such as genotyping errors or manageable population stratification [85].

In case-control studies, P-value correction is essential to control Type I error (false positive). The goal is to ensure that the probability of finding at least one false positive in the entire set of tests (known as the Family-Wise Error Rate - FWER) or the expected proportion of false positives among significant tests (False Discovery Rate - FDR) is maintained at an acceptable level [86]. While FWER control provides strong protection against errors, FDR control offers a more powerful alternative, making it highly suitable for studies involving multiple hypotheses where maximizing true discoveries is a

priority [87]. Although fundamental in association studies, only Alves Pereira Neto et al. [47] reported performing p-value adjustments (Bonferroni method) for the SNVs in the *CD209* and *VDR* genes. While the absence of multiple testing correction does not inherently invalidate the reported findings, it necessitates cautious interpretation of isolated significant results. In this context, the present systematic review plays a central role by critically organizing the evidence, highlighting that part of the reported genetic associations may reflect statistical artefacts rather than robust biological effects. Future studies should prioritize adequate correction for multiple comparisons, replication in independent cohorts, and harmonized analytical strategies to improve the reliability and interpretability of genetic association signals in PCM.

The independent reanalysis of genetic association data constitutes a critical step for verifying the robustness and reproducibility of reported findings. By reconstructing the analytical pipeline, it allows for the validation of quality control procedures, statistical methods, and the application of appropriate multiple testing corrections, thereby safeguarding against spurious associations arising from technical artifacts or analytical flexibility [88]. Furthermore, reanalysis enables sensitivity analyses under different genetic models and covariate adjustments, ensuring that results are not contingent on a single analytical approach [89]. We conducted a data reanalysis using SNPstats to rigorously address two fundamental methodological challenges in genetic association meta-analyses: (i) the evaluation of Hardy-Weinberg equilibrium and (ii) the statistical determination of the optimal genetic model for each variant [90].

Therefore, upon reanalyzing the data from the seven studies identified in our review, we found that in the study by Bozzi et al. [48], which investigated the association between PCM and variations in the *IL10* gene, the best-fitting inheritance model yielded an Odds Ratio result 2.3 times lower than originally reported. Conversely, in the study by Alves Pereira Neto et al. [47], reanalysis showed a 1.2-fold increase in the odds ratio for the association between PCM and VDR SNVs, with the recessive model continuing to provide the best fit according to AIC and BIC values (S1 Table). Thus, the reanalysis process is indispensable for confirming the validity of genetic signals, exploring potential biases such as population stratification, and ultimately, strengthening the evidence base before proceeding with costly replication efforts in independent cohorts [91].

**Geographical distribution of the case and control groups.** Lastly, an important factor to be considered in association studies is that the control group should be collected in the same region as the case group. Of the seven articles selected in this review, only the work of Lozano et al. [52] used patients from three Brazilian states (Goiás, Minas Gerais, and São Paulo) while the control group consisted of samples from only one state (Goiás). These authors did not specify the geographical origin of the groups in the article, as the information was obtained via personal communication. Another important point regarding the origin of the case and control groups is found in the work of Mendonça et al. [53]. The authors did not specify which region each group belongs to; they only mention that the study was conducted at two Brazilian centers: the General Hospital of the Federal University of Triângulo Mineiro (UFTM) and the University Hospital of Botucatu Medical School, São Paulo State University (UNESP). Thus, it is not possible to know if the groups are formed by a mixture from both locations or if each group comes from a specific region. Unfortunately, we were unable to access this information through direct communication with the authors. The remaining studies included in this review used case and control groups from the same region, with Bozzi et al. [48,49] and Alves Pereira Neto et al. [47] using samples from the state of Minas Gerais, and Carvalho et al. [50] and Sato et al. [55] from the state of São Paulo.

However, even with all the limitations mentioned above, the Hardy-Weinberg equilibrium test was performed in the majority of studies, and no deviation was found (Table 1). This suggests that the case-control sample was drawn from a general population unaffected by natural selection, migration, mutation, or random drift. It is important to mention that, according to a recent study that addressed Brazilian genetic diversity [92], although the three states mentioned in the studies (Goiás, Minas Gerais, and São Paulo) share a predominantly European genetic profile and similar history of miscegenation, there are notable differences in the proportion of African and indigenous ancestry, as well as local variations due to historical and geographical factors. Goiás, for example, has a more distinct genetic signature, with greater indigenous influence and less recent diversity, while São Paulo and Minas Gerais are more similar to each other, but still

with nuances (such as the greater African contribution in Minas Gerais). Thus, it is important to note that when conducting genetic association studies, it is important to take into account not only the geographical distribution of the case and control groups, but mainly the origin and genetic ancestry that populations may present.

**Exposure measurements.** The selection of appropriate control groups is a critical aspect of genetic association studies, particularly in the context of infectious diseases, where prior exposure to the pathogen must be carefully considered. In studies on PCM, the most suitable controls are individuals who have lived and worked for extended periods in rural areas with known risk of exposure [6]. In this context, a positive intradermal test using *Paracoccidioides*-specific antigens in clinically healthy individuals represents an important criterion for control selection, as it reliably indicates previous fungal exposure [93]. Failure to adequately account for exposure status may result in exposure misclassification, which can bias effect estimates such as odds ratios in case-control studies [94]. Therefore, rigorous and consistent assessment of exposure is essential to ensure the validity and interpretability of genetic association findings.

In our review, we found that only Mendonça et al. [53] used the skin test for the detection of paracoccidioidic infection. Regarding intradermal tests, it should be noted that the use of paracoccidioidin and other antigens pose health risks due to the lack of quality and standardization of the majority of these immunobiologicals. Thus, following the publication of Ordinance No. 686 by the Ministry of Health, Health Surveillance Service, on August 27, 1998, and until immunobiological production centers comply with technical standards for safety and quality control, the Brazilian National Health Surveillance Agency discontinued the recommendation for the use of antigenic preparations, including paracoccidioidin, for the evaluation of delayed-type hypersensitivity reactions. Even so, cutaneous tests with antigens of the *P. brasiliensis* are devoid of diagnostic value but are very useful in establishing the prognosis and in the follow-up of the patients [6,94,95].

In addition, in both studies in which significant associations between SNVs and PCM were identified after reanalysis, the authors did not report the adoption of adequate measures to assess pathogen exposure [47,48]. Consequently, the interpretation of these findings and the conclusions drawn from them should be made with caution, as the observed associations may be affected by exposure-related bias.

## Recommendations for Future Research

PCM is a complex disease that emerges from the interaction between host susceptibility and fungal pathogenicity. Current clinical evidence in human PCM does not consistently support species-specific differences in disease presentation or severity [7]. As a result, most association studies in humans have historically focused on host genetic and immunological factors that modulate susceptibility and clinical outcomes. Importantly, this approach reflects limitations in available clinical data rather than an assumption of fungal homogeneity. Moreover, the reliance on a limited number of laboratory-adapted isolates, such as *P. brasiliensis* Pb18 [43,44], represents an additional limitation when extrapolating experimental findings to natural infections. Prolonged in vitro cultivation of pathogenic fungi can drive laboratory adaptation, leading to alterations in the expression of virulence-associated traits and, in some cases, to attenuated phenotypes. These adaptations may affect cell wall architecture, enzyme secretion, host–pathogen interactions, and stress response pathways, thereby potentially distorting conclusions about fungal behavior in vivo [95,96]. Future studies integrating fungal genomic diversity, strain-level phenotyping, and host genetic data in well-characterized clinical cohorts will be essential to fully elucidate the contribution of pathogen variability to PCM pathogenesis.

To facilitate future work involving meta-analyses of genetic association, we recommend that upcoming studies primarily address the following points: (i) The genotypes of all component studies should be tested for Hardy-Weinberg equilibrium and excluded if a significant deviation is observed [97]. (ii) Since most tested variants are unlikely to be genuinely associated with the phenotype of interest, random false-positive signals may occur and obscure true associations. To prevent this, multiple-testing correction methods, particularly the False Discovery Rate, must be applied to adjust the p-values of

odds ratios and reduce the likelihood of spurious findings [98]. (iii) Achieving sufficient statistical power to detect significant associations requires a relatively large sample size, which for certain diseases, such as PCM, is nearly impossible to obtain individually. To address this, meta-analyses or replication studies with step-wise approaches can be employed as a means of increasing the effective sample size. However, it is essential that individual-level data from each study be made available, particularly genotypes and the characterization of covariates such as age, sex, ethnicity, and comorbidities [99]. Covariate-adjusted analyses should additionally be performed to minimize biases arising from differences between cases and controls.

## Conclusion

This systematic review underscores a complex genetic landscape in PCM, identifying significant associations between susceptibility to the disease and variants in key immune-related genes, namely Interleukin 10 and the Vitamin D Receptor. However, the interpretation of their conclusions should be approached with caution. Nonetheless, the failure of the meta-analysis to corroborate these associations underscores a critical finding: the current evidence base is markedly constrained by methodological limitations. These constraints include underpowered sample sizes, deviations from Hardy-Weinberg Equilibrium, a prevalent lack of correction for multiple testing, inadequate matching of controls for geographical and genetic ancestry, and the absence of covariate-adjusted analyses. A pivotal insight from our independent reanalysis was that reported associations are highly sensitive to the chosen statistical model. Consequently, elucidating the true genetic underpinnings of PCM will necessitate future studies with enhanced methodological stringency, prioritizing larger, well-matched cohorts, rigorous statistical adjustments, and, most importantly, full data transparency to facilitate conclusive meta-analyses and independent validation.

## Supporting information

**S1 Table. Association of single nucleotide variants with the presence of paracoccidioidomycosis (PCM), according to the studies retrieved in the systematic review.** The table presents the genetic models tested (codominant, dominant, recessive, overdominant, and log-additive), the genotype frequencies in healthy individuals and PCM patients, the odds ratios (OR) with 95% confidence intervals, p-values (including FDR correction), as well as the information criteria (AIC and BIC). Deviations from Hardy–Weinberg equilibrium (HWE) are highlighted in bold and in blue. Significant associations are highlighted in bold and in red (susceptibility). The best inheritance model, according to AIC and/or BIC, is highlighted in yellow. *CTLA4*: cytotoxic T-lymphocyte associated protein 4; *CD209*: CD209 molecule; *FCGR2A*: Fc gamma receptor IIa; *IFNG*: interferon gamma; *IL10*: interleukin 10; *IL12A*: interleukin 12A; *IL12B*: interleukin 12B; *IL12RB1*: interleukin 12 receptor subunit beta 1; *IL18*: interleukin 18; *IL4*: interleukin 4; *JAK1*: Janus kinase 1; *TNF*: tumor necrosis factor; *VDR*: vitamin D receptor.
(XLSX)

**S2 Table. Association of single nucleotide variants with the presence of paracoccidioidomycosis (PCM), according to the meta-analysis of the studies retrieved in the systematic review.** The table presents the genetic models tested (codominant, dominant, recessive, overdominant, and log-additive), the genotype frequencies in healthy individuals and PCM patients, the odds ratios (OR) with 95% confidence intervals, p-values (including FDR correction), as well as the information criteria (AIC and BIC). Deviations from Hardy–Weinberg equilibrium (HWE) are highlighted in bold and in red. *IFNG*: interferon gamma; *IL4*: interleukin 4; *TNF*: tumor necrosis factor.
(XLSX)

**S3 Table. Summary of studies included in the systematic review evaluating the association between single nucleotide variants and paracoccidioidomycosis.** The table compiles key information on study design such as data source, genes investigated, local and sample size of each group.
(XLSX)

**S1 PRISMA Checklist. Completed PRISMA 2020 checklist for systematic reviews.** From: Page MJ, McKenzie JE, Bossuyt PM, Boutron I, Hoffmann TC, Mulrow CD, et al. The PRISMA 2020 statement: an updated guideline for reporting systematic reviews. BMJ 2021;372:n71. https://doi.org/10.1136/bmj.n71.
(DOCX)

## Author contributions

**Conceptualization:** Wellington Santos Fava, Ana Carla Pereira-Latini, Alessandra Pontillo, James Venturini.

**Data curation:** Sanderson da Silva Coelho, Wellington Santos Fava.

**Formal analysis:** Sanderson da Silva Coelho, Wellington Santos Fava, Ana Carla Pereira-Latini, Alessandra Pontillo.

**Investigation:** Sanderson da Silva Coelho, Wellington Santos Fava, Eva Burger, Ana Carla Pereira-Latini, Alessandra Pontillo, James Venturini.

**Methodology:** Wellington Santos Fava, Eva Burger, Ana Carla Pereira-Latini, Alessandra Pontillo.

**Project administration:** James Venturini.

**Resources:** James Venturini.

**Software:** Wellington Santos Fava.

**Supervision:** James Venturini.

**Validation:** Wellington Santos Fava.

**Visualization:** Eva Burger, James Venturini.

**Writing – original draft:** Sanderson da Silva Coelho, Wellington Santos Fava, Eva Burger, Ana Carla Pereira-Latini, Alessandra Pontillo, James Venturini.

**Writing – review & editing:** Sanderson da Silva Coelho, Wellington Santos Fava, Eva Burger, Ana Carla Pereira-Latini, Alessandra Pontillo, James Venturini.

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
