## [Decision Letter · Decision Letter 0]

26 Dec 2025

PNTD-D-25-02109

Genetic background and immune response in paracoccidioidomycosis: a systematic review and meta-analysis of single nucleotide variants

Dear Dr. Venturini,

Thank you for submitting your manuscript to PLOS Neglected Tropical Diseases. After careful consideration, we feel that it has merit but does not fully meet PLOS Neglected Tropical Diseases's publication criteria as it currently stands. Therefore, we invite you to submit a revised version of the manuscript that addresses the points raised during the review process.

Please submit your revised manuscript within by Feb 24 2026 11:59PM. If you will need more time than this to complete your revisions, please reply to this message or contact the journal office at plosntds@plos.org. Please include the following items when submitting your revised manuscript:

We look forward to receiving your revised manuscript.

Kind regards,

Joshua Nosanchuk, MD

Section Editor

Shaden Kamhawi

co-Editor-in-Chief

Paul Brindley

co-Editor-in-Chief

**Journal Requirements:**

1) We noticed that you used “unpublished data" in the manuscript. We do not allow these references, as the PLOS data access policy requires that all data be either published with the manuscript or made available in a publicly accessible database. Please amend the supplementary material to include the referenced data or remove the references.

2) Figure 2: please (a) provide a direct link to the base layer of the map (i.e., the country or region border shape) and ensure this is also included in the figure legend; and (b) provide a link to the terms of use / license information for the base layer image or shapefile. We cannot publish proprietary or copyrighted maps (e.g. Google Maps, Mapquest) and the terms of use for your map base layer must be compatible with our CC BY 4.0 license.

**Reviewers' Comments:**

Reviewer's Responses to Questions

**Key Review Criteria Required for Acceptance?**

**Methods:**

-Are the objectives of the study clearly articulated with a clear testable hypothesis stated?

-Is the study design appropriate to address the stated objectives?

-Is the population clearly described and appropriate for the hypothesis being tested?

-Is the sample size sufficient to ensure adequate power to address the hypothesis being tested?

-Were correct statistical analysis used to support conclusions?

-Are there concerns about ethical or regulatory requirements being met?

Reviewer #1: (No Response)

Reviewer #2: The study presents clearly defined objectives and the methodology used is appropriate.

**Results**

-Does the analysis presented match the analysis plan?

-Are the results clearly and completely presented?

-Are the figures (Tables, Images) of sufficient quality for clarity?

Reviewer #1: (No Response)

Reviewer #2: The analyses performed are generally consistent with the proposed analysis plan; however, the manuscript has limitations in data analysis that were not adequately addressed or discussed. The suggestions and notes have been included in the section "Summary and General Comments".

**Conclusions**

-Are the conclusions supported by the data presented?

-Are the limitations of analysis clearly described?

-Do the authors discuss how these data can be helpful to advance our understanding of the topic under study?

-Is public health relevance addressed?

Reviewer #1: (No Response)

Reviewer #2: The conclusions are generally supported by the data presented. The manuscript acknowledges some limitations of the analyses; however, additional limitations should be more clearly addressed and discussed.

**Editorial and Data Presentation Modifications?**

Reviewer #1: (No Response)

Reviewer #2: (No Response)

**Summary and General Comments**

Reviewer #1: The manuscript by Coelho et al., addresses an important topic: the genetic basis of susceptibility to paracoccidioidomycosis (PCM), a neglected tropical disease endemic to Latin America. The review and meta-analysis are relevant. However, the study faces methodological limitations that affect the robustness of its conclusions.

Concerns:

The meta-analysis includes only two studies per SNV, which severely limits statistical power. This should be emphasized more clearly in the abstract and discussion.

Funnel plot asymmetry tests were omitted due to insufficient studies, this limitation should be stated in the methods and discussion.

Only one study applied multiple testing correction. The absence of FDR or Bonferroni adjustments in most studies increases the risk of false positives. This limitation should be more prominently addressed.

The discussion is somewhat repetitive. It also should separate methodological limitations from biological interpretation.

Figure captions must be improved to make them self-explanatory.

Reviewer #2: This manuscript by Coelho and colleagues addresses an important and underexplored topic. It looks at how host genetic differences affect susceptibility and clinical outcomes in paracoccidioidomycosis (PCM), which is a neglected disease relevant to public health in Latin America. The authors conducted a systematic review and reanalysis of published studies that focus on single nucleotide variants (SNVs) in immune-related genes and carried out a limited meta-analysis. Their effort to critically reassess published data, including Hardy-Weinberg equilibrium (HWE), inheritance models, and effect sizes, is commendable and meets the standards expected by this journal. However, despite its strengths, the manuscript has some problems that significantly limit its impact and may bias the conclusions. The main issues are discussed below.

Major concerns

1. A central limitation of the manuscript is the assumption that PCM susceptibility and severity are driven mainly by host genetic and immunological factors. While the introduction and discussion emphasize host immune responses, such as the Th1/Th2 balance, cytokine polymorphisms, PRRs, and VDR signaling, they largely overlook the role of the fungal pathogen itself. PCM is defined by host-pathogen interaction. There are well-documented differences in virulence, immunomodulatory capacity, thermotolerance, antigen expression, and cell wall composition among Paracoccidioides species and strains, such as P. brasiliensis and P. lutzii. These differences have significant effects on disease outcomes. The manuscript fails to adequately discuss how fungal genetic diversity, virulence factors, or strain-specific traits may influence host genetic associations. This omission risks presenting an oversimplified and potentially misleading view that PCM is “primarily a host-driven disease.” In reality, disease development stems from a dynamic interaction between host susceptibility and fungal pathogenicity. Additionally, it should be noted that the studies cited in references 7, 8, 9, 10, and 11 exclusively used the P. brasiliensis Pb18 isolate. Prolonged in vitro cultivation of pathogenic fungi is known to lead to significant changes in the expression of virulence-related components, often resulting in attenuated phenotypes. This laboratory adaptation can affect cell wall composition, secretion of enzymes, host-pathogen interactions, and stress response pathways. Therefore, relying on a single isolate that may have undergone changes due to long-term culture presents an additional limitation of these studies. This should be considered when interpreting conclusions related to fungal virulence and host-pathogen interactions. The authors should acknowledge this limitation and frame their conclusions within a host-pathogen interaction model instead of a host-centric view.

2. Though the manuscript correctly highlights associations involving IL10 and VDR SNVs, the biological interpretation remains shallow. Most of the variants mentioned are found in promoter, intronic, or untranslated regions, yet there is little discussion regarding whether these SNVs have any functional impact, such as altered cytokine production, receptor expression, or signaling efficiency in PCM or related fungal infections.

3. Another significant weakness is the inconsistency in defining “exposure” and “controls” across studies. Only two studies included controls with confirmed exposure to Paracoccidioides, while the others depended on apparently healthy individuals whose exposure status was unknown. Given the high prevalence of subclinical infection in endemic areas, this poses a critical confounding factor. The manuscript notes that none of the significant associations were convincingly tied to clinical form or disease severity, but it doesn’t integrate this finding into its conclusions effectively. As a result, the conclusions about “susceptibility” are vague and may be confused with exposure, infection, and progression.

Minor concerns

a) The introduction focuses heavily on immunology and could benefit from a brief section that recognizes fungal diversity and virulence as important factors in PCM.

b) Some repetition occurs in the Discussion concerning HWE and multiple testing corrections; this could be streamlined.

c) The authors might consider including a reformulated version of Supplementary Table 1S, retaining its main points of the table, in the main text, as this could improve clarity and strengthen the presentation of the results.

PLOS authors have the option to publish the peer review history of their article (what does this mean? ). If published, this will include your full peer review and any attached files.

**Do you want your identity to be public for this peer review?** For information about this choice, including consent withdrawal, please see our Privacy Policy .

Reviewer #1: No

Reviewer #2: No

**Figure resubmission:**
---

## [Editor Report · Decision Letter 1]

16 Feb 2026

PNTD-D-25-02109R1

Genetic background and immune response in paracoccidioidomycosis: a systematic review and meta-analysis of single nucleotide variants

Dear Dr. Venturini,

Thank you for submitting your manuscript to PLOS Neglected Tropical Diseases. Although it is clear that thoughtful revisions have been made by the authors, the response to reviewers document does not provide a point by point response to individual comments from the two prior reviewers, which is required in order to effectively assess whether the changes in the revision are sufficient to address ALL the issues raised during review. Therefore, we invite you to submit a revised version of the manuscript that addresses the points raised during the review process.

Please submit your revised manuscript within by Mar 18 2026 11:59PM. If you will need more time than this to complete your revisions, please reply to this message or contact the journal office at plosntds@plos.org. Please include the following items when submitting your revised manuscript:

We look forward to receiving your revised manuscript.

Kind regards,

Joshua Nosanchuk

Section Editor

Shaden Kamhawi

co-Editor-in-Chief

Paul Brindley

co-Editor-in-Chief

**Reviewers' Comments:**

Need to provide formal point by point responses to reviewer comments to the original submission.

**Figure resubmission:**
---

## [Editor Report · Decision Letter 2]

3 Mar 2026

Dear Dr. Venturini,

The authors are applauded for their rigorous response to the feedback on their work. We are pleased to inform you that your manuscript 'Genetic background and immune response in paracoccidioidomycosis: a systematic review and meta-analysis of single nucleotide variants' has been provisionally accepted for publication in PLOS Neglected Tropical Diseases.

Best regards,

Joshua Nosanchuk

Section Editor

Shaden Kamhawi

co-Editor-in-Chief

Paul Brindley

co-Editor-in-Chief
